# A real-time feedback system stabilises the regulation of worker reproduction under various colony sizes

Simeon Adejumo[1]*, Tomonori Kikuchi[2], Kazuki Tsuji[3], Kana Maruyama-Onda[4], Ken Sugawara[5], Yoshikatsu Hayashi[1]¤*

1 Biomedical Sciences and Biomedical Engineering, School of Biological Sciences, University of Reading, Reading, Berkshire, United Kingdom, 2 Marine Biosystems Research Center, Chiba University, Tokawa 1, Choshi City, Chiba, Japan, 3 Department of Subtropical Agro-Environmental Sciences, University of the Ryukyus, Nishihara, Okinawa, Japan, 4 Nagoya Protect Station, Shimizu Sub-Station, Ministry of Agriculture, Forestry and Fisheries, Shizuoka, Japan, 5 Tohoku Gakuin University, 2-1-1, Tenjinzawa, Izumi-ku, Sendai, Japan

¤ Current address: Biomedical Sciences and Biomedical Engineering, School of Biological Sciences, University of Reading, Reading, Berkshire, United Kingdom
* s.adejumo@pgr.reading.ac.uk (SA); y.hayashi@reading.ac.uk (YH)

**Data Availability Statement:** All data and code used is available at Zenodo (DOI: 10.5281/zenodo.6759932).

## Abstract

Social insects demonstrate adaptive behaviour for a given colony size. Remarkably, most species do this even without visual information in a dark environment. However, how they achieve this is yet unknown. Based on individual trait expression, an agent-based simulation was used to identify an explicit mechanism for understanding colony size dependent behaviour. Through repeated physical contact between the queen and individual workers, individual colony members monitor the physiological states of others, reflecting such contact information in their physiology and behaviour. Feedback between the sensing of physiological states and the corresponding behaviour patterns leads to self-organisation with colonies shifting according to their size. We showed (1) the queen can exhibit adaptive behaviour patterns for the increase in colony size while density per space remains unchanged, and (2) such physical constraints can underlie the adaptive switching of colony stages from successful patrol behaviour to unsuccessful patrol behaviour, which leads to constant ovary development (production of reproductive castes). The feedback loops embedded in the queen between the perception of internal states of the workers and behavioural patterns can explain the adaptive behaviour as a function of colony size.

## Author summary

In the ant *Diacamma cf. Indicum* (from Japan), the queen spends more effort on queen pheromone-transmitting behaviour (patrolling) in response to the growth of colony size to inhibit worker ovary development. We used an agent-based simulation to understand the mechanism of the colony size dependent behaviour of the queen. The queen simply follows a feedback loop mediated by the mutual contact between her and the workers. In

**Funding:** The author(s) received no specific funding for this work.

**Competing interests:** The authors have declared that no competing interests exist.

other words, the queen patrols the workers more often when she has recently encountered workers with developed ovaries.

We found that this self-regulatory mechanism works even when the worker density per space was kept constant. We also found that despite the presence of such feedback, the effectiveness of the queen patrol, and thus, the suppression of worker ovarian activity decreased with increasing colony size. This indicates that a colonial phase shift from the ergonomic stage to the reproductive stage, a general phenomenon in social insect colonies, emerged as the colony grew.

## Introduction

Various collective behaviours in social insects are regulated by self-organisation [1]. While many studies have addressed questions about how short-term (finite) collective behaviour (foraging, moving, etc., dynamics from the beginning to the end of behaviour) is autonomously controlled, there is relatively scant knowledge about the homeostatic mechanisms of societies. Namely, while the homeostatic mechanisms of individual organisms, such as breathing, thermoregulation, and osmoregulation have been thoroughly studied, how society is autonomously maintained has attracted less attention.

Each species or population of social insects may have a characteristic value in social traits such as colony size and caste ratios. However, the colony size can dramatically change over time as the colony grows (like in the body size of multicellular organisms [2]), and the caste composition and age composition also can show much shorter-term fluctuations [3–5]. The reproductive division of labour through suppression of worker reproduction, which is a hallmark of insect eusociality, is maintained even during such changes in the internal environment of the colony. This implies the existence of a size-free autonomous control mechanism, but their detailed mechanisms have not been clarified [6].

From a phylogenetic point of view, physical suppression via dominance behaviour is considered an ancestral state of insect eusociality and has changed to chemical suppression using the queen pheromone [7] as the colony size increases. The reason is likely due to a physical constraint since direct interference would be difficult if the colony size increased above a certain point. However, control problems could still arise as the colony size increases. There are also control problems which could arise during the process of colony development (colonial ontogeny) where physical suppression is still practical as a form of controlling worker ovary development.

So far, mechanisms of the reproductive division of labour, such as dominance behaviour, queen pheromones and worker policing, have been extensively studied [8–10], but less about the robustness and effectiveness of the regulation mechanisms against the increase in colony size [11–14].

Based on a computer simulation model, we examined how robust the regulation system for worker reproduction is when colony size changes. Workers in many Hymenoptera are prevented from laying eggs in the presence of the queen, instead engaging in non-reproductive work. For workers, this reproductive altruism is considered to be an adaptive tactic in terms of inclusive fitness optimisation [15, 16].

In many species, when the queen dies or becomes absent for any reason, the worker's ovaries begin to develop and eventually lay male-destined haploid eggs. This switch in reproduction is triggered by the perception of the queen's presence. Therefore, the transmission of

information on the existence of the queen is the key to understanding the mechanism of the reproductive division of labour [10, 17–19].

Information about the queen's existence has been considered to be transmitted by a chemical substance (queen pheromone). Empirical studies in recent years revealed that the main body of the queen pheromone is low volatility cuticular hydrocarbons (CHCs) that are conserved widely in social Hymenopteran taxa such as ants, bees and wasps [20]. Analogous solutions has also been found in termites in the form of 9-ODA [21, 22]. In an environment where the queen pheromone has low volatility, its transmission is thought to require direct physical contact between the queen and her nest mates.

In this study, we focused on *Diacamma* ants as a model system [23]. Specifically the Japanese *Diacamma*, *Diacamma.cf. Indicum*, the only Japanese species. The information transmission mechanism of the gamergate worker of this species (known henceforth as the queen) has been well established [24]. This information is coded by CHCs and transmitted by direct contact between the queen and workers [25]. The queen is reported to exhibit specific behaviours to improve the information transmission efficiency of her presence. The locomotory activity of the ant queen in the nest is generally not as high as that of workers, but in the Japanese *Diacamma*, the queen frequently roams the nest (this is called patrol [26]).

Importantly, in large colonies, the queen's patrol is more active than that in small colonies [24]. In other words, the queen is buffering the possible decrease in the transmission efficiency (contact probability) of the queen pheromone to workers. Since in *Diacamma* the physiological effect (suppression of ovarian development of workers) of the queen pheromone can last only 3 hours or so [24], stable control of worker reproduction seems to require the perception of ever-changing colony size, thereby the queen can adjust her patrol effort. However, a question remains about how the queen obtains colony size information and reflects it in her actions.

In ants, the frequency of contacts between individuals is a local population-size proxy, and such contact frequency is used for behavioural switching in various contexts such as moving the nest [27, 28]. Note, however, that the fundamental mechanisms which can link different types of perception such as the frequency of contacts, local density perception and colony size perception have been largely unclear.

In general, the feedback mechanism is essential for system stability and has been identified in various collective actions created by self-organisation [29–31]. In *Diacamma*, it is known that the queen acts aggressively towards reproductive workers and when she encounters an egg-laying worker during her patrol, she steals and destroys the egg (queen policing) [32]. Therefore, we assume the queen can detect the reproductive status of the worker when she contacts it during her patrol. There is at least some circumstantial evidence that the queen can detect the reproductive state of workers from Shimoji *et al.* [33]. The queen can suppress worker reproduction through dominance interactions. The ability to sense worker reproductive status is a necessary precursor to determine if and when these dominance interactions should take place.

Previous work by Sugawara *et al.* [34] theoretically suggested that a feedback mechanism could play a role in the colony size dependant patrol of *Diacamma* queens. Their model had three assumptions: (1) A worker that has lost contact with the queen for a significant period is released from the inhibitory effects of the queen pheromone; (2) such a worker starts ovarian development and also starts herself emission of the queen pheromone (or other chemicals associated with ovarian development); (3) when the queen, on patrol, comes into contact with a worker who emits such chemicals, the queen increases her future patrol effort according to the chemical concentration she perceives.

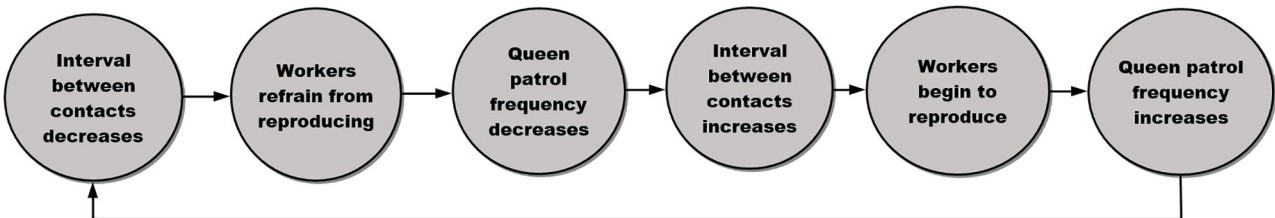

**Fig 1. Feedback loop diagram.** A feedback loop between the queen's patrol behaviour and the reproductive activity of workers.

As the colony size increases, the contact efficiency of the queen decreases, and the workers have more chances to develop their ovaries and emit the associated chemicals. The colony-size-dependent behaviour of the queen would be a result of feedback mechanisms in response to changes in the worker's physiological condition (Fig 1).

Sugawara *et al.*'s model assumed that the queen contacts the workers at a constant rate (number of contacts per unit time), and as the colony grows, the contact rate with the queen per worker decreases linearly, increasing the patrol frequency of the queen (Fig 1). It is, however, not self-evident that the queen and workers have a constant contact rate. For instance, workers' behaviour may change as the density of workers in a given space increases. This itself is thought to be a response to changes in contact frequency to reduce the probability of contact between individuals [35].

However, the average individual density for a given space in the ant's nest may not change very much, even if the colony size changes. Franks *et al.* [36] found that ants changed the size of their nest space to fit with the colony size, making changes to the nest as required. Though others [37] found that the increase in nest space is slower than the increase in colony size, the largest variation in worker density occurred due to seasonal changes. Inter-individual distance also appears to be regulated [38]. Indeed in *Diacamma* the worker density per space in the nest remains almost constant (see the experimental results in Appendix 4, S4 Fig in appendix).

As the previous method by Sugawara *et al.* did not include the spatial aspect, our approach to this problem is to use an agent-based simulation in which the queen and workers can move around and interact in a defined space. Regardless of the colony size, the worker density per space (surface area) was assumed to remain constant. Thus, the queen was simulated under the condition that the absolute contact rate with other individuals (workers) would not be an indicator of colony size. The mathematical model by Sugawara *et al.* [34] focused only on the queen's patrol and did not investigate changes in worker reproductive status. We need to keep in mind that queen pheromones suppress worker reproduction, so the queen needs to evaluate the internal state of the workers. That is, if activated, the patrol behaviour of the queen results in the inactivation of the workers' ovaries, suppressing the development of ovaries. Furthermore, since the contact between the queen and the workers is a stochastic event, the contact interval tends to vary from individual to individual. Therefore, it is unclear whether a simple increase in patrol time or frequency is effective in limiting the reproduction of workers at a colony-wide level.

This is the first time that an agent-based has been used to investigate the effectiveness of the queen's patrol behaviour via the tracking of the internal state of workers. Although simulations have been used in other biological contexts such as Boids. Boids are an agent-based simulation [39] that replicates the complex flight patterns observed in groups of flocking birds by simulating the interaction between the individuals of the flock. The agent-based simulation can also be used for predicting the behaviour of the flock given simple rules the individuals follow.

However, the Boids do not use feedback between individual agents to determine the flight path of the flock, instead, monitoring their neighbours.

Previous studies were also able to successfully simulate the nest quality assessment behaviour of ants. These include Şahin and Franks [40], which utilised a free mobile robot simulator to study nest assessment dynamics in a similar way to the current paper. Similar approaches were used by Perna *et al.* [31], Marshall *et al.* [41], and Shiraishi *et al.* [42] which were able to reproduce results observed in the literature for trail formation. For the agent-based simulation proposed in this paper, the agents have an internal state which modulates the behaviour of the agents and reacts to the mutual interactions which occur between agents to form a negative feedback loop. This leads to control over the division of reproductive labour in the colony.

Using an agent-based simulation has substantial advantages in studying our hypothesis of a negative feedback mechanism. As a result, it is possible to program an individual organism's behaviour and its interactions with other organisms with certain degrees of freedom, such as density. In other words, group behaviour arises as a result of the contact between agents and between the agents and their environment, rather than attempting to represent system-level phenomena [43, 44]. Also, it gives other advantages such as allowing a closer examination of how the internal dynamics which characterise individual behaviour are coupled with the more complex group behaviour [45, 46].

In this study, using an agent-based simulation, we investigated how the reproductive state (ovarian development) of the workers and the patrol behaviour of the queen are affected by the change in colony size. We set the condition that the individual density per nest space is constant even if the colony size changes. We also provide experimental evidence and results supporting the constant-density assumption and validating simulation results. We propose a feedback mechanism between the internal states and the patrol behaviour of the queen. We discuss how the feedback mechanism contributes to the stable suppression of worker reproduction as the colony size increases.

## Materials and methods

### Maintenance and experimental procedure

The taxonomic status of species of genus *Diacamma* is still under revision. Since it is known that there are only one species of this genus in Japan that is very closely related to the Indian species *Diacamma Indicum*, we tentatively use the new name *Diacamma cf. Indicum* (from Japan) following Fujioka *et al.* [47](previously described as *Diacamma sp.* from Japan). The species has no morphological castes among females, that is, all females are wingless and monomorphic. In each colony, a single mated female (queen) functions as the reproductive queen that produces female eggs, whereas the other females play the helper-worker role [48].

New colonies are founded via colony fission. When the queen is absent after fission or due to queen mortality, among the cohort of newly emerged females the most dominant individual (usually the first emerged) copulates and becomes the next queen. In the field, colonies contain 20–300 workers, and alates (males) are produced in large queen-right colonies and orphan colonies [49, 50]. Unmated workers can potentially lay male-destined haploid eggs. However, in colonies at the ergonomic (growing) stage (i.e., ones with fewer than 100 workers), worker reproduction is suppressed by queen pheromone and, queen and worker policing [25, 32, 50, 51]. Whereas in colonies at the reproductive stage (containing 100 workers or more) such suppression is relaxed, and males are produced by worker reproduction [50].

We used colonies of *Diacamma cf. indicum* collected on the main island of Okinawa during 2001–2014. Those colonies were maintained in a laboratory at 25 ±1˚C with a light: dark cycle of 12 h:12 h. Each colony was kept in a plastic container (26.5 cm length × 18.5 cm width × 5

cm height) with a plaster floor (1.5 cm thick), however, in the natural environment, ants would explore the surrounding environments to expand the colony space whenever there is an opportunity. In the middle of the floor, a $13 \times 9$ cm depression (1 cm deep) covered with a glass plate was prepared for the ants as an artificial nest. Ants were fed honey water and mealworms ad libitum three or four times a week.

Ants were kept in the laboratory. First, all workers and the queen in each of the 15 colonies were marked with enamel paint for individual identification. The number of workers (colony size) was 58, 69, 81, 110, 125, 128, 131, 144, 149, 151, 162, 169, 174, 181, and 214, respectively (mean ± SD = 137.9 ± 42.7). For the video recording, each colony was moved to another artificial nest, which was a plastic container (26.5 cm length × 18.5 cm width × 5 cm height) with a plaster floor (1.5 cm thick). In the middle of the floor, a depression (8 cm length × 16 cm width × 1 cm depth) covered with a glass plate was prepared for the ants as an artificial nest. After acclimatisation for a day, we video-recorded each colony for 12h. By using that video data, we were able to track all queen–worker contacts.

## Agent-based simulations

**Overview.** To validate our hypothesis of the negative feedback loop between the queen and workers, we ran the agent-based simulations in which the queen and workers move randomly within the grid space and contact each other. The internal state of the workers is defined as the hypothetical physiological condition, such as the hormone level, which controls the ovary development in workers and queen pheromone secretion in the queen. Within an ant colony, the queen's perception of the internal state via contact is largely dependent on the frequency of her contacts with workers as a function of time. Thus, along with the internal dynamics of the queen and workers, the spatial distribution of the workers and the queen as a function of time should play an important role in the patrol behaviour of queens and the reproductive behaviour of workers. Note that the density of workers was kept the same though the number of workers (colony size) increased. This can distinguish the mechanism based on the negative feedback loop from those dependent on the perception of density [35]. To include the spatial degree of freedom, we used an agent-based simulation to model the behaviour of the queen and individual workers within a certain space representing the nest.

We first assumed that the internal state of the workers and the queen would operate differently. For the worker, the internal state would represent their ovary development and demonstrate a steady increase over time. This could be suppressed by the queen via direct contact (perception of the queen pheromone). For the queen, the internal state would represent the probability to become active. That is, the likelihood that the queen will go from an inactive state to an active state, at which point she will begin to patrol the colony. The queen's internal state steadily decreases (increasing her resting period) but increases when interacting with workers. This increase is proportional to the internal state of the worker who has been contacted. Meaning, a worker with a low internal state has minimal effect on the queen's internal state but a worker with a high internal state increases the queen's internal state and therefore her likelihood to begin patrolling the colony.

The queen's movement [52, 53] around the nest was based on her temporal behaviour: when she is in the active state, she moves around the space, whereas in the inactive state, she halts within the grid she had moved in. The contact behaviour of the queen depended on these temporal behavioural patterns of active–inactive cycles. In this case, the workers also have active-inactive cycles which determined their movement around the space and were determined a priori.

The movement of the queen and workers around the colony was a simple random walk around the nest space. The next position of the agents is generated randomly from one of 4 directions, North, South, East and West. The movements of the agents are asynchronous, with agents only moving when they are in an active state. The queen walks around the nest space to contact all workers in the colony to suppress their internal states. The duration of the simulation was determined by how long it took the queen to contact all workers at least once. The simulation was then terminated. Various variables were recorded for analysis, including the active and inactive period of the queen and the contacts between the queen and workers.

**Internal state dynamics.** The rhythmic cycle of the active–inactive state was simplified into the two time periods of the active time ($t_a$) and the inactive time ($t_r$). For the worker agents, $t_a$ and $t_r$ were kept constant ($t_{a_{constant}} = 20$ steps, $t_{r_{constant}} = 100$ steps). For the queen agent, $t_a$ was kept constant ($t_{a_{constant}} = 20$ steps), but $t_r(t)$ was modulated by her internal state, $I_q(t)$, using the dynamics of:

$$t_r(t) = t_{r_{constant}} \cdot e^{-\delta \cdot I_q(t)} \tag{1}$$

where $t_{r_{constant}}$ and $\delta$ were constant ($t_{r_{constant}} = 100$ steps, $\delta = 20.0$). $I_q(t)$ represents the likelihood that the queen will transition to patrolling at time t. Increases in $I_q(t)$ lead to a decrease in the inactive time, $t_r(t)$, of the queen. If the queen has a prolonged period where her internal state is low, then $t_r(t)$ approaches $t_{r_{constant}}$.

To test our hypothesis about the feedback mechanisms of internal state and behaviour, we model the dynamics of the internal states of the queen and workers in the following manner. The dynamics of the internal state of the queen is given by:

$$I_q(t+1) = (1-\epsilon) \cdot I_q(t) + \alpha \cdot \delta(\overrightarrow{x_q}, \overrightarrow{x_w}) \cdot I_w(t) \tag{2}$$

$(1-\epsilon) \cdot I_q(t)$ is a damping factor, where $\epsilon = 0.01$ and is constant. As the reproductive division of labour enables the queen to be the main producer of offspring in the colony, there is a compromise between patrolling the colony and laying eggs. As the internal state of the queen represents the likelihood she will become active, we use the damping factor to naturally decrease the queen's internal state over time. This allows the queen to move to a more restful state, where there is minimal patrol, assuming workers have a low internal state. The second term, $\alpha \cdot \delta(\overrightarrow{x_q}, \overrightarrow{x_w}) \cdot I_w(t)$, is an activation factor. The activation factor increases the probability of the queen becoming active when the queen contacts a worker with a high internal state. It is also proportional to the number of contacts with the workers. $\alpha = 0.1$ and is constant. $\overrightarrow{x_g}$ and $\overrightarrow{x_w}$ denote the position of the queen and worker respectively. The term $\delta(r)$ denotes Kronecker's delta, i.e., its value is zero except when the distance between $\overrightarrow{x_q}$ and $\overrightarrow{x_w}$ is zero, then $\delta(r) = 1$.

The dynamics of the internal state of the workers was given by:

$$I_w(t+1) = (1-\beta) \cdot I_w(t) + \gamma - \kappa \cdot \delta(\overrightarrow{x_q}, \overrightarrow{x_w}) \cdot I_w(t) \tag{3}$$

$\gamma$ and $\beta$ represent an activation factor that increases the internal state of the worker over time as a function of time. This levels off over time as $I_w(t)$ approaches 1. In this case, $\gamma$ and $\beta$ were set to 0.0001. Here we chose the constants of the activation factor to reflect the pace of ovary development in workers observed in previous work where the queen was removed from the colony [24]. The next term only functions to decrease the internal state of the worker, $I_w(t)$, when the worker is contacted by the queen with $\kappa = 0.9009$, representing an approximately 90% decrease in the worker's internal state.

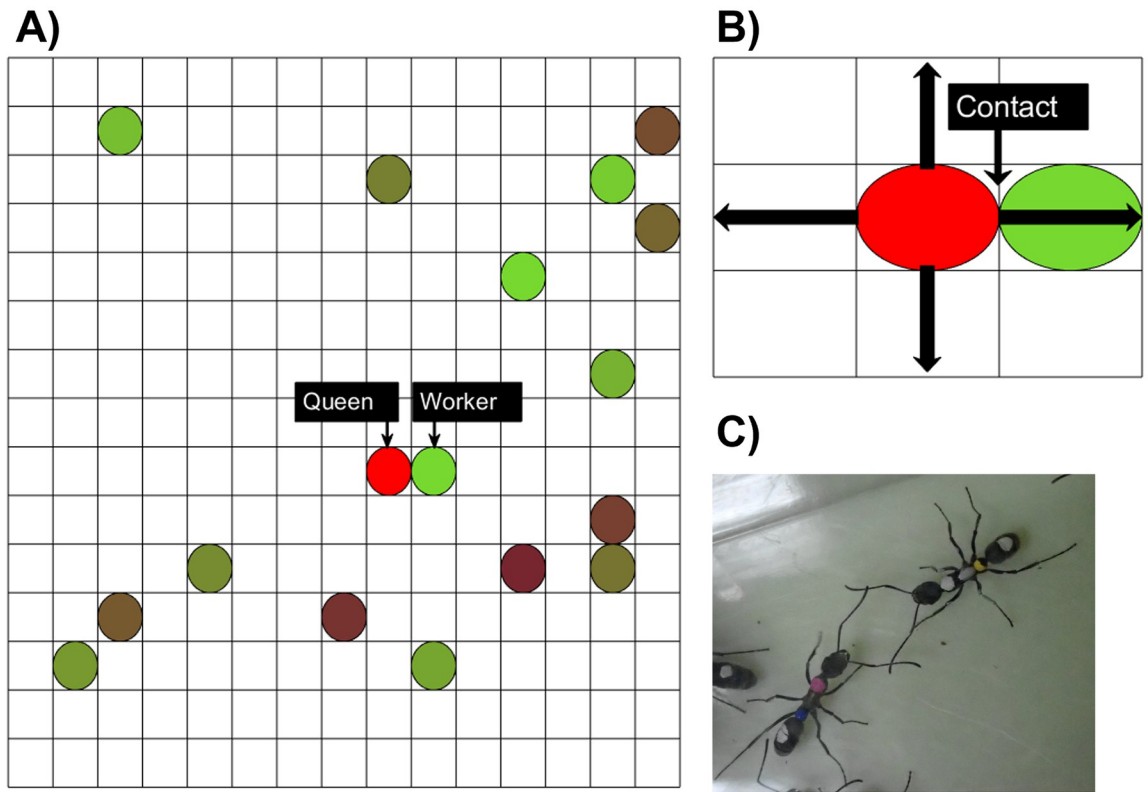

**Fig 2. Schematic picture of the simulation environment.** A) Grid structure with the queen and workers. B) Contact is established when workers are within range of the queen. In this instance, the threshold was set to 5 units of length. C) Two ants contacting each other by touching antennae.

**Spatial behavioural dynamics.** While we understand that the movement of ants in a real colony are less than random, for simplicity, we implemented spatial dynamics in the following way:

1. The virtual nest was set with a grid size of $L \times L$ (Fig 2A) with distance measured in arbitrary units (simply referred to as units). The ants (agents) were distributed randomly throughout the nest space at the start of the simulation. The size of the nest space was dependent on the colony size to keep the density ($N/L^2$) approximately constant ($L$ was set proportional to the square root of $N$). For example, when $N = 20$, $L = 100$ units. We controlled $L$ to keep the ant density per space constant (this assumption was based on empirical evidence as shown in Appendix 3 and Appendix 4).

2. Each agent was set to be 5 units long. Every time step the agents move randomly in one of four directions: north, south, east, or west (Fig 2B) in the grid. The agents are prevented from going outside of the virtual nest space with a simple check of their next position vs the position of the boundaries of the space.

3. Agents are unable to overlap each other within the single grid.

**4.** A contact is determined when a worker is close to the queen, within the length of 5 units (see evidence of the ant's morphology Fig 2C). The queen can only contact one worker in each time step. Therefore we decided that the queen would not contact the same worker twice in a row. This is to decrease the prospect of a worker who has already been contacted recently monopolising contact with the queen despite other workers being in range in a short period of time. When the queen contacts the worker, the internal state of the queen and the worker increase and decrease respectively. The increase in the queen's internal state is proportional to the internal state of the worker, while the decrease in the worker's internal state is constant (approx. a 90% decrease).

**Initial conditions and analysis.**   Conditions of workers and the queen were initialised with parameters that represent the position, direction, velocity and internal state. The internal states of the workers were randomly assigned a value between 0 and 0.5. The queen was given an initial internal state of 0.1. The number of ant workers, $N$, was predefined to sample the different colony sizes. The initial rest time of the queen is set to the maximum rest time ($t_{r_{constant}} = 100$). The status of the workers and queen (whether it is active or inactive) were randomly assigned at the beginning of the simulation. Each agent, either a worker or the queen, has its internal state, $I_w$ and $I_q$, respectively. $I_w$, the internal state, is assumed to decrease when the worker contacts the queen but to increase otherwise (Eq 3). $I_q$ is assumed to increase when the queen encounters a worker with high $I_w$ and to decrease in the absence of such an encounter (Eq 2). The queen's internal state is assumed to be correlated with her patrol behaviour, i.e., a higher $I_q$ leads to a shorter resting time, $t_r$.

The simulation ended when the queen had contacted all the workers in the colony at least once. The simulation was repeated 50 times for each of the colony sizes $N = 20-200$ increasing $N$ in increments of 20. The colony size coincides with the range of natural *Diacamma cf. Indicum* colony sizes [24]. In every trial, the positions of the workers are reset to another random value (different initial conditions for spatial distributions of workers). The total time of the simulation, patrol frequency and length of the rest time were recorded. The internal states of the workers and the contacts between agents were recorded. Using these variables, the effect of the queen's patrol behaviour could be analysed by calculating the average internal state of workers over time, as well as the distribution of these internal states. Contact rates between the queen and workers were also calculated based on the number of contacts made between the queen and the workers within the simulation time. All variables used in the simulation are shown in Table 1.

We tested the robustness of our model in several ways. Firstly, we checked the initialisation of parameters. By increasing the initial internal of the queen, we assessed its effect on the system. We found that it had little affect and returned to similar values seen in our original initialisation (see Appendix 8, S8 and S9 Figs). We then standardised the time across the colony sizes which were investigated to mitigate possible effects due to the system not being in a steady state. We also found this to have little effect (possibly strengthening our results, see Appendix 9, S10 Fig). Finally, we increased $\beta$ and $\gamma$ to 10x their original values. While we found a significant difference in the results (see Appendix 10, S11 Fig), the dynamics of the system were unchanged.

## Results

In the experimental results, overall in small colonies with fewer than 100 workers, the queen was able to contact more than 80% of workers in the 20 bouts of patrols, whereas, in large colonies with more than 100 workers, the queens' per worker contact frequency dramatically

**Table 1. Variables, constants, and initial conditions used in the agent-based simulation.**

| Variable Symbol | Variable Name | Value |
|---|---|---|
| $L$ | Grid length and width | 100(when N = 20) |
| $N$ | Number of workers (colony size) | 20,40,60,80,100,120,140,160,180,200 |
| $t_r(t=0)$ | Resting time for the queen | Randomly assigned between 0 and $t_{r_{constant}}$ |
| $t_{r_{constant}}$ | Maximum resting time | 100 |
| $t_{a_{constant}}$ | Maximum active time | 20 |
| $\delta$ | Delta(constant) | 20 |
| $\epsilon$ | Epsilon(constant) | 0.01 |
| $\alpha$ | Alpha(constant) | 0.1 |
| $\gamma$ | Gamma(constant) | 0.0001 |
| $\beta$ | Beta(constant) | 0.0001 |
| $\kappa$ | Kappa(constant) | 0.9009 |
| $I_q(t=0)$ | Queen internal state | 0.1 |
| $I_w(t=0)$ | Worker internal state | Randomly assigned between 0 and 0.5 |
| $\overrightarrow{x_q}$ | Queen position | Randomly assigned between 0 and L |
| $\overrightarrow{x_w}$ | Worker position | Randomly assigned between 0 and L |

decreased (Fig 3). These results suggest that although the queen increased her patrol effort with increasing colony size, the efficiency of making contacts between the queen and workers dropped in the large colonies.

The first set of simulation results displays colony size dependent features of the queen's patrol behaviour. Fig 4 showed the patrol frequency and rest time of the queen with respect to

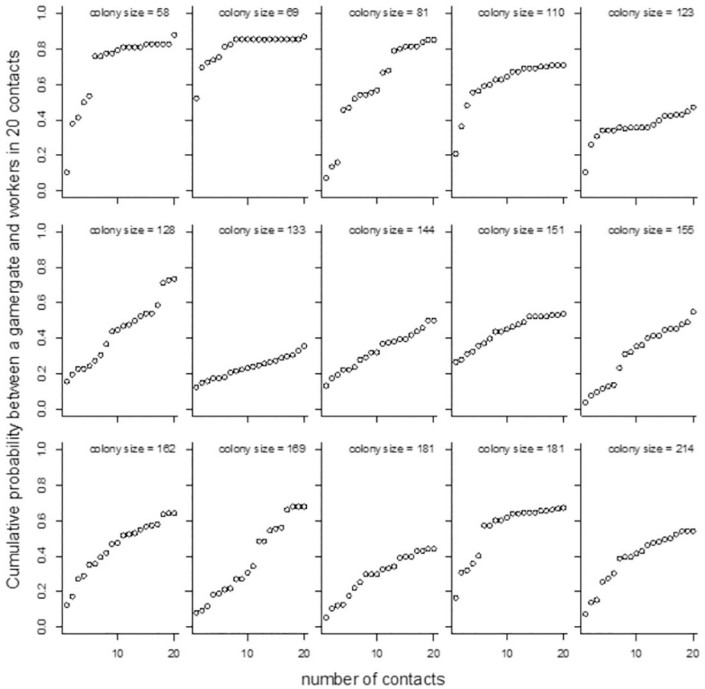

**Fig 3. Cumulative proportion of workers contacted by the queen during patrols at various colony sizes during laboratory experiments.**

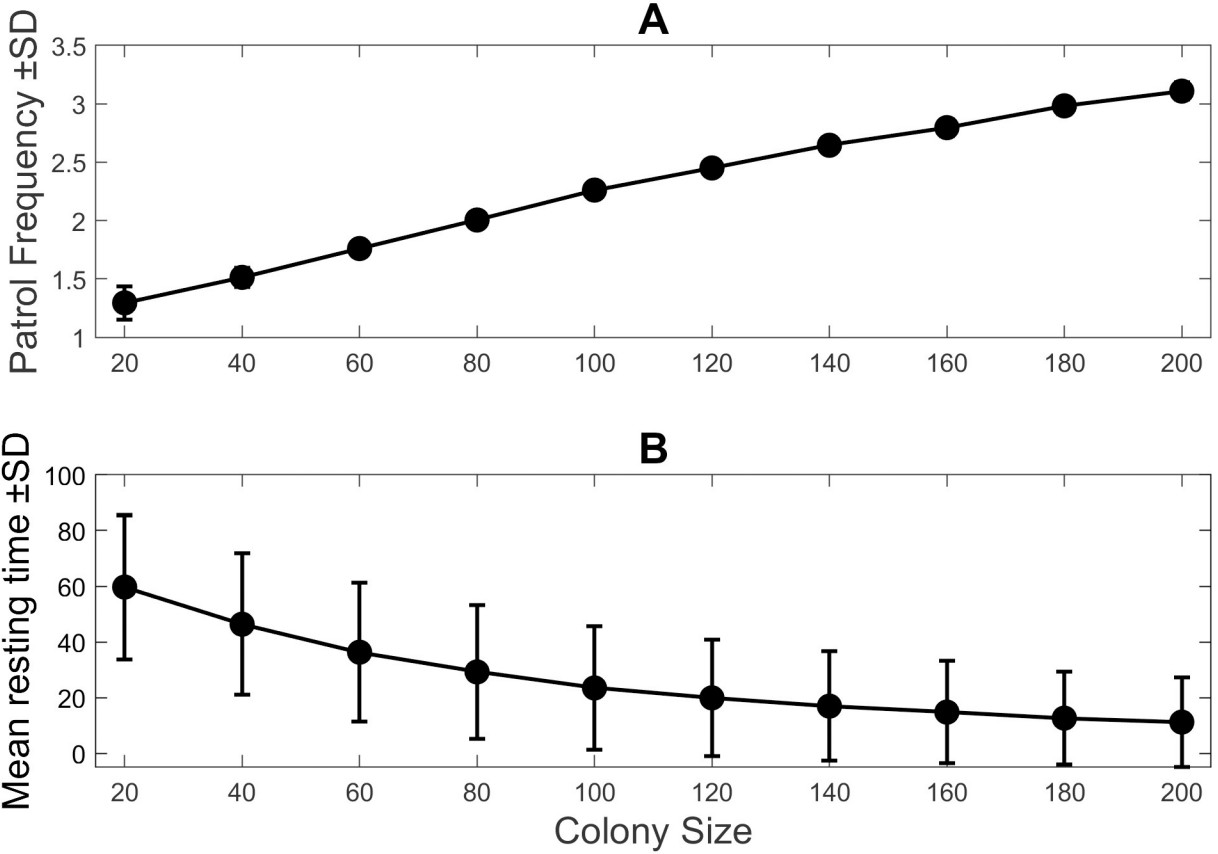

**Fig 4. Patrol frequency and encounters.** (A) Patrol frequency of the queen as a function of the colony size. The patrol frequency of the queen increases as the colony size increases. This shows an increase in patrol effort by the queen. (B) Resting time as a function of the colony size shows an inverse relationship to the patrol frequency. The mean resting time decreases with colony size as the queen spends more time patrolling

colony size. In this case, the patrol frequency refers to how often the queen patrols the colony at a given colony size. The rest time is the total amount of time the queen spends inactive for a given colony size. Fig 4A showed an increase in the patrol frequency of the queen with respect to colony size (see also Appendix 6, S6 Fig) due to the increased internal state of workers. Inversely, Fig 4B showed that the mean resting time for the queen decreased with colony size, i.e., as the colony size increases, the queen increases her patrol effort to contact an increasing number of workers in the colony. The increase in patrol frequency does not lead to constant patrolling by the queen at large colony sizes, which would be impossible for a real queen due to physical restrictions.

These results qualitatively agreed with the experimental data reported by Kikuchi *et al.* [24]. Kikuchi *et al.*, through colony size manipulation, also showed that the queen increased her patrol effort with increasing colony size. This was also confirmed through our own experiments (Appendix 1, S1 Fig). As a next step, let us determine the effectiveness of the queen's patrol behaviour in controlling the internal state of workers in the colony.

To determine the effectiveness of the queen's patrol behaviour, distributions of worker internal states were calculated over time. Fig 5 shows the distributions of the internal states of workers (when *N* = 20), comparing two cases where real-time feedback was implemented and not implemented (contact with the queen had no impact on worker state as a control). Here,

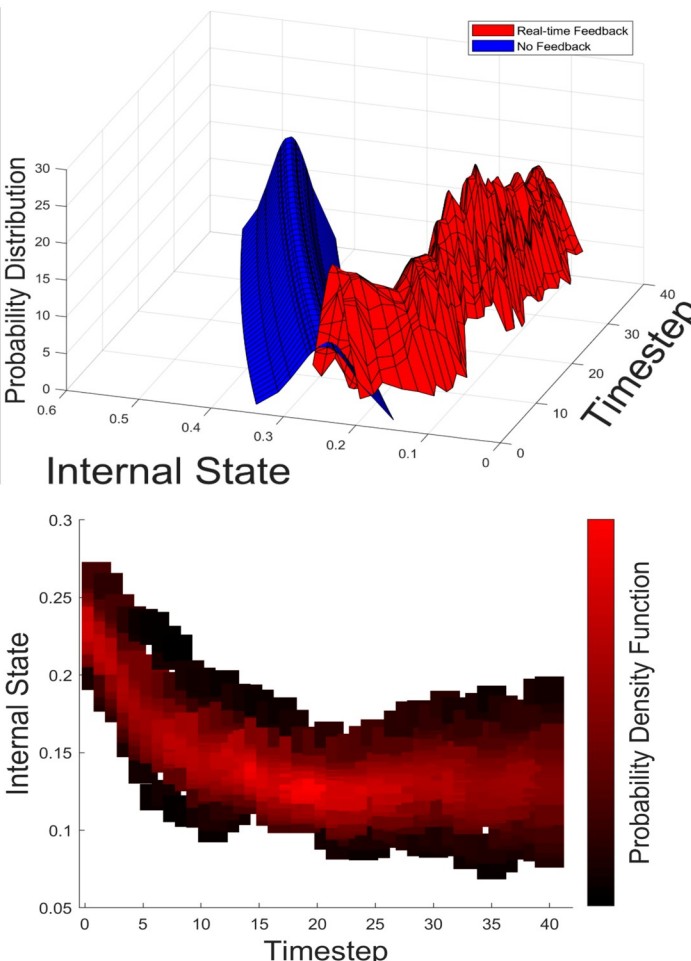

**Fig 5. Probability distribution of worker's internal state (N = 20).** (Top) Probability distribution of workers' internal states over time in a colony of 20 workers with (red) and without (blue) real-time feedback. We compared the results of simulations with real-time feedback and with a no-feedback case to study how queen patrol behaviour suppresses the internal state of workers. No feedback (blue) causes continuous growth of the internal states regardless of the colony size. With real-time feedback in relatively small colonies, the average value of the workers' internal states decreases from 0.2361 to 0.1344. (Bottom) This can be seen more clearly in the bottom plot, with the decrease in the internal state of workers from the initial value. The intensity of the colour shows the probability density function, with a larger proportion of workers being close to the mean. The shift of the distribution for real-time feedback demonstrates that the patrol behaviour is successful in suppressing the internal states of workers.

we could quantify the internal states as a function of time in the agent-based simulations, which cannot be obtained experimentally.

These results indicate that the real-time feedback model of the queen's patrol behaviour suppresses the internal states of the workers effectively with smaller variance than the case when there is no feedback. Results for a larger colony size, $N = 120$, show that the feedback model can be effective in controlling the internal state of workers when compared to no the feedback case (S5 Fig). However, there appears to be an increase in the mean internal state and the variance of the distribution. Therefore, as colony size increases, the effectiveness of the queen's patrol behaviour decreases.

The decrease in the efficiency of the queen's patrol behaviour can be shown more clearly in Fig 6 which shows the mean internal states of workers over time for different colony sizes

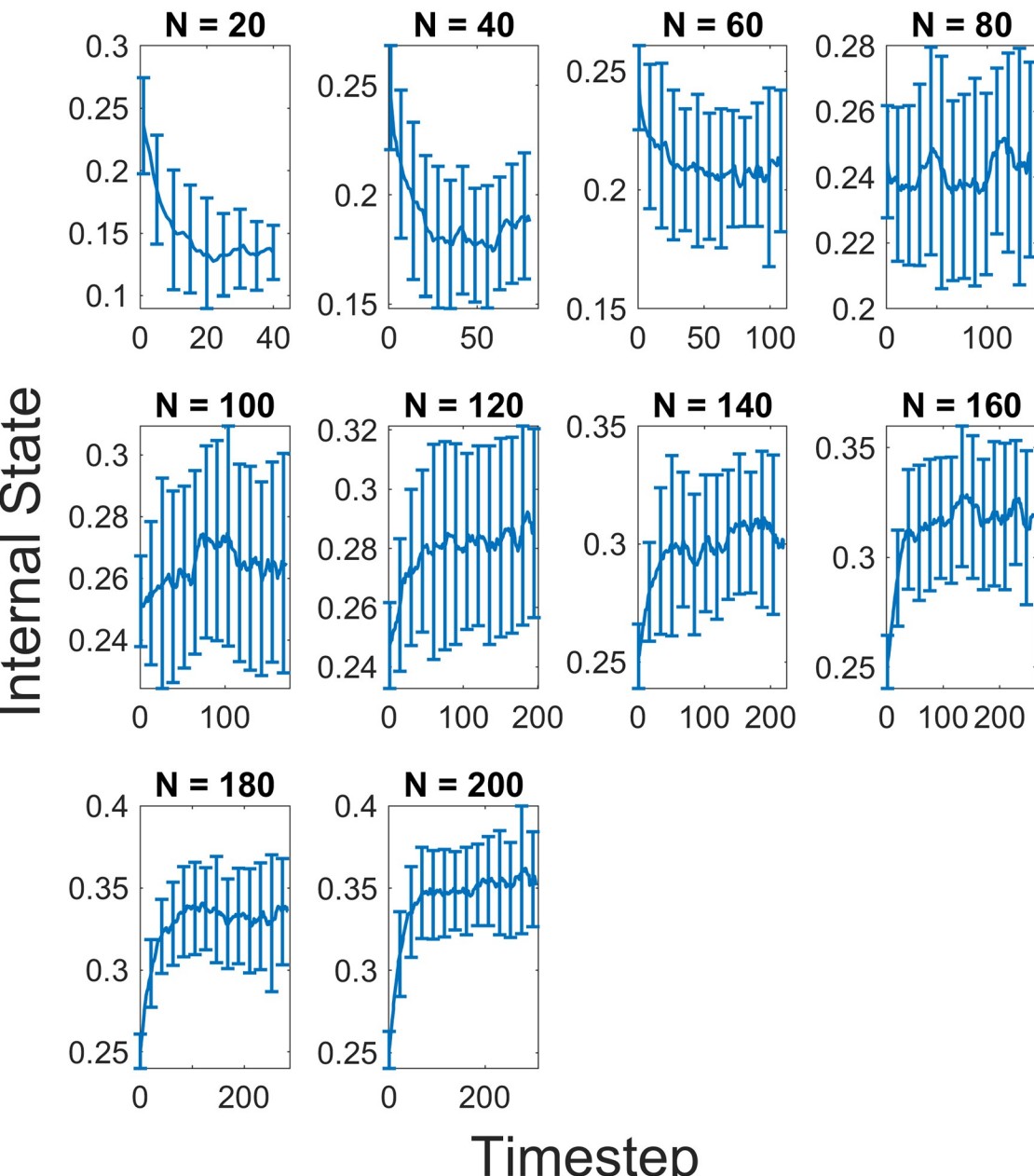

**Fig 6. Average internal state of the colony for various colony sizes ($N$ = 20–200 at increments of 20 workers).** As the internal states of workers were initialised randomly between 0 and 0.5, the average initial value for all colony sizes is approximately 0.25. At smaller colony sizes ($N$ = 20, 40, and 60), the average internal state of workers decreases as a function of time and stabilises at a point which is lower than the initial value. As the colony size becomes larger, there appears to be a transition (at $N$ = 80 and 100) where there is greater fluctuation in the average values over time. At the largest colony sizes ($N$ = 140 to 200), there is an increase in the average internal state of workers over time. Here too there is stabilisation but at a point higher than the initial value.

ranging from $N$ = 20 to $N$ = 200. In smaller colony sizes, we observe a decrease in the mean internal state of workers as a function of time from the initial random values of the internal states.

This confirms that the suppression of the internal states was realised sufficiently via physical contact by the queen. This indicates that the feedback loops between the perception of the

internal states and the decrease of the rest time in patrol worked effectively. Also, in the spatial degree of freedom, the queen (through her random walk) was able to contact all the workers who were also walking around randomly even though the colony size increases.

When the colony size increases further, there appears to be an inflexion point, between $N = 80$ and $N = 100$ (Fig 6), where the mean internal state begins to increase rather than decrease as a function of time. This shows a decrease in the effectiveness of the queen's suppression of worker internal states or the start of the failure of the patrol behaviour. This can be seen more clearly in the larger colony sizes ($N = 120$ to $N = 200$). The lack of suppression at this stage is due to the contact between the queen and workers and not the physical limitation of the queen. While there is an increase in the mean internal state with colony size, there appears to be relative stabilisation in the mean after some time. As a next step, let us interrogate the mechanism in the spatial degree of freedom, namely the number of contacts from the queen to the workers, and vice versa.

By logging the number of contacts that occurred during the simulation, various contact rates could be calculated. These are the queen contact rate, the per-worker contact rate and the contact rate between workers. The contact rate is defined as the number of contacts per unit time. Therefore, the queen contact rate is the rate at which the queen contacts workers per unit time. While the per-worker contact rate is the average contact rate of a worker in the colony. The contact rate between workers is the rate workers contact any other worker in the colony. These values were calculated separately, with contacts logged for the queen and individual workers. Theoretically, assuming an even distribution of contacts between workers, the per-worker contact rate is equivalent to the queen contact rate divided by the number of workers. However, the per-worker contact rate conveys the contact efficiency of the queen and, therefore, the effectiveness of the queen's patrol behaviour.

Fig 7 shows various contact rates between the queen and workers as a function of the colony size. Fig 7A shows the overall contact rate for the queen increasing (black line) while the per-worker contact rate decreases (blue line). Despite the increased patrol effort by the queen (shown in Fig 4), the contact efficiency of the queen decreases with colony size. This is due to the insufficient increase in the queen contact rate. Distinctions were made for the contact rate during the rest cycles (Fig 7B) and patrol cycles (Fig 7C) of the queen. This was to demonstrate that the majority of the contacts by the queen were made when the queen was patrolling.

Note here that all the results obtained in the agent-based simulations were predicated on constant density. The results so far indicate that while the queen contacts more workers in larger colonies, based on more frequent patrols, the lower contact per worker leads to an increase in the mean internal state of workers in the colony due to the decrease in contact efficiency.

The loss of contact efficiency may be due to a colony size dependent effect on the patrol behaviour of the queen. This should not affect how workers contact each other. To test this, we quantified the contact rate between workers, shown in Fig 8. Similar to Fig 7A, the contact rate between workers increases as a function of colony size, but the per-worker contact rate between workers decreases. Note again that the results were obtained based on constant density. The per-worker contact rate decreases for the same reason it decreases for the queen. The contact efficiency is lost at larger colonies because the agent's movement is insufficient to cover the space. However, for the queen, we can relate this to her behaviour because her internal state is related to her movement.

In comparison, the worker's movement stays the same at all times. Hence, the loss in per-worker contact rate between workers is likely more significant than the drop in per-worker contact rate with the queen. Although there are more workers than the queen and as the simulation ends once the queen has contacted every worker at least once, there is a more

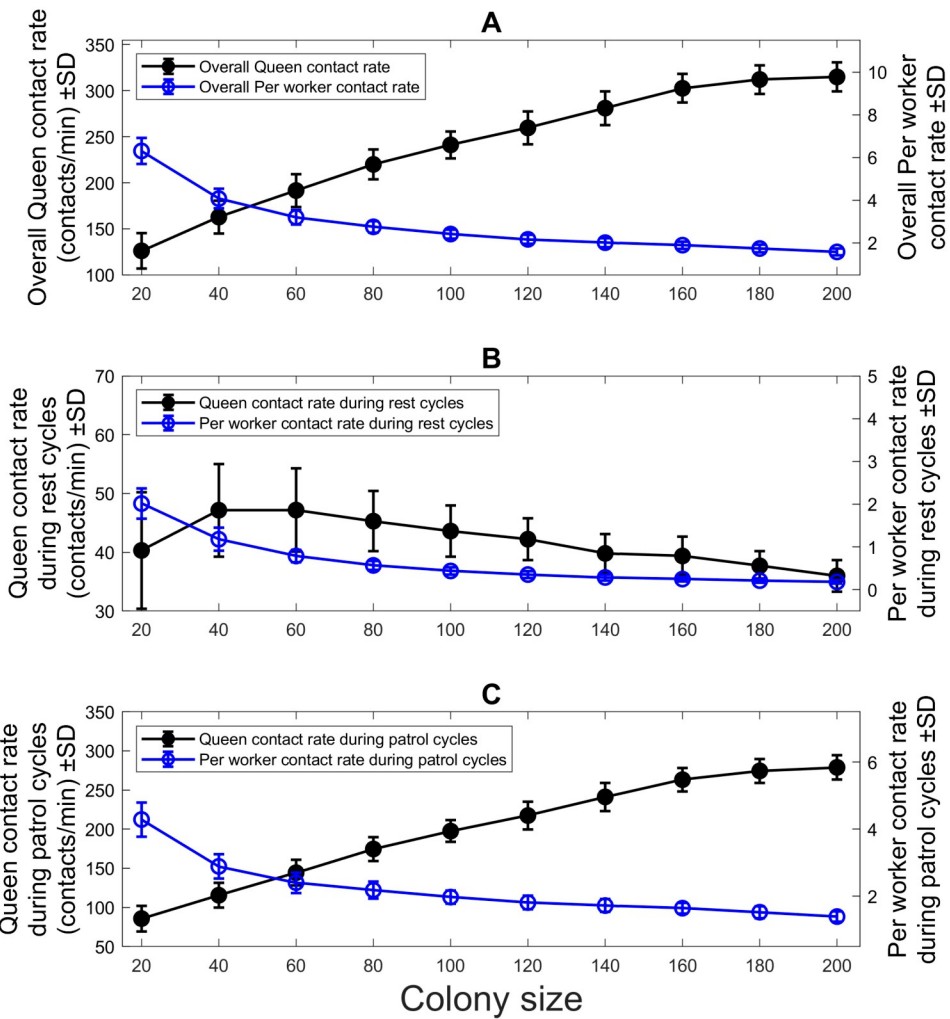

**Fig 7. Contact rates between the queen and workers.** (A) The overall queen contact rate and the per-worker contact rate. The overall queen contact rate increases with colony size. This reflects an increase in the patrol effort as well as the increase in colony size. However, the per-worker contact rate decreases with colony size, showing a decrease in the contact efficiency of the queen. (B) The queen contact rate and per worker contact rate during the rest cycles of the queen. While there is a slight increase in the queen contact rate, overall the trend is a decrease in both the queen and per-worker contact rate with colony size. This means leads to (C) The queen contact rate and per worker contact rate during the patrol cycles of the queen. There is an increase in the queen contact rate, showing that more workers are contacted during the patrol cycles of the queen than during the rest cycles. However, there is still a decrease in the per-worker contact rate, similar to Fig 7A & 7B.

considerable drop in the per-worker contact rate with the queen in Fig 7A than in Fig 8. The workers do not have to have unique contact with other workers. In contrast, the queen does have unique contacts because she needs to address every worker individually.

## Discussion

Our results from the agent-based simulations revealed that the real-time feedback system between a queen and workers can have an influential role in maintaining and stabilising the internal states of the workers under various colony sizes. The simulations showed that, with a constant density, the queen increased her patrol frequency as the colony size grew (Fig 4A), and as a result, she could suppress the internal states of workers effectively (Fig 5). The

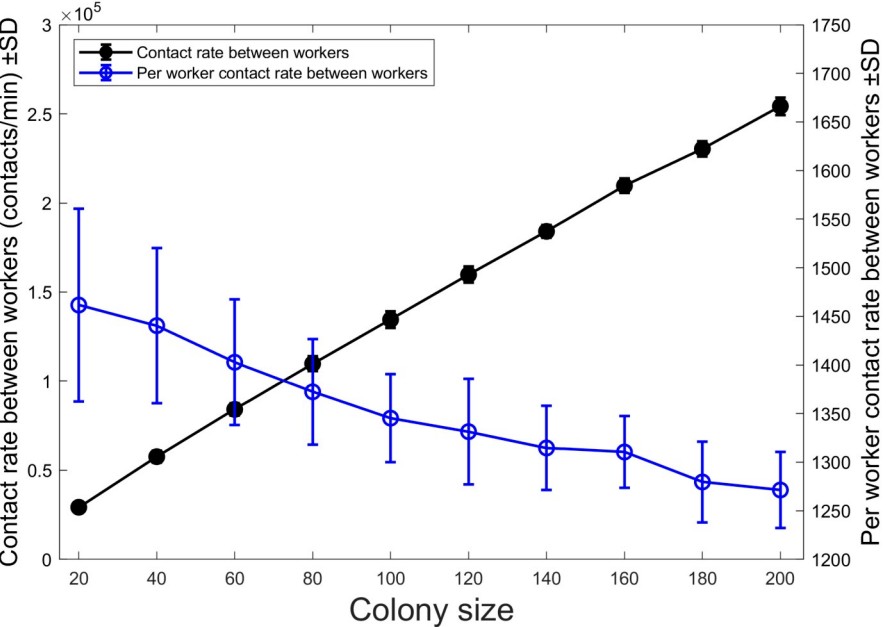

**Fig 8. Contact rate between workers.** The contact rate between workers (black line) increases with colony size. An increased colony means more workers so there will be more contacts in general. When looking at the per-worker contact rate (blue line), there is a decrease with colony size similar to that seen in Fig 7A. There appears to be a decrease in the contact efficiency not just between the queen and workers but also between workers.

underlying feedback mechanism is as follows: When the average internal state of workers increases, the queen frequently perceives a larger internal state, leading to an increase in the queen's internal state (2). This increase in the queen's internal state in turn leads to an increase in the queen's patrol frequency by decreasing her resting time (1). In short, as the colony size increases, the per-worker contact rate (Fig 7A, blue line) decreases, which triggers an increase in the queen's patrol frequency (Fig 4A). Hence, the queen's sterility-maintaining behaviour in response to an increasing colony size is revealed. However, this was only the case until the colony reached certain colony sizes. In larger colonies, $N = 120$ to $N = 200$, the queen contact efficiency became low (Fig 7A), and consequently, the internal states of workers were no longer effectively suppressed, i.e., the average internal states increased as a function of time (Fig 6) and at the end of the run many workers were ready to perform self-reproduction.

This simulation result was qualitatively consistent with what was observed in real *Diacamma* colonies. Namely, a positive association between the queen patrol effort and colony size (Appendix 1, S1 Fig, see also [24]), and the effective suppression of worker reproduction in small colonies and less effective suppression in larger colonies [49, 50]. The feedback loops through physical contact between queens and workers are sufficient to suppress the internal state of workers in small colonies (Fig 5). In theory, such colony size dependent worker reproduction is adaptive in terms of the inclusive fitness of workers in monogynous and monandrous hymenopteran colonies [16]. Suppression of worker's reproduction when the colony is small (ergonomic stage) contributes to rapid colony growth. When the colony is large (reproductive stage), worker-produced eggs are less policed and more likely to survive [50], which can imply that the selfish option (worker reproduction) may benefit workers.

We are the first to explicitly state a hypothetical proximate mechanism generating the colony size dependent character expression and the shift from ergonomic to reproductive stages. This is a general pattern in social insects.

More importantly, both the reproductive division of labour among a queen and workers and the switch in the colony stages (from ergonomic to reproductive) are simply achieved by the decision making of member individuals who just rely on personally acquired local information of recently encountered individuals. Decentralised control and self-organisation are thought to be the mechanisms that give rise to various adaptive functions of social insect colonies, such as the allocation of the workforce to various tasks that the colony needs, and selective recruitment of foragers to better food sources among the food sources available [54–56]. These theories commonly argue that single colony members have access to only limited "local" information, but they perform adaptively as a whole [57, 58]. So far, the "overall" frequency of encounters with other individuals related to local density in a nest has been often discussed as a piece of effective colony-size information for each colony member to decide their behaviour [28, 35]. However, in this study, we assumed that the individual density per nest space, thus the contact frequency with other individuals per time per individual, is constant even if the colony size changes. We consider that in real ants a positive correlation of individual density per space with colony size can occur. This can occur in situations in which ants have physical difficulty in expanding their own nest space. However, in the absence of such a spatial constraint, it would be more natural to assume that ants extend the housing architecture of the nest as the colony grows. For this reason, we consider that local density, or the simple frequency of encounters, does not generally serve as reliable information on total colony size. Actually, in *Diacamma* (Appendix 4, S4 Fig) individual density per nest space is likely regulated to be more or less constant. Also, in some ants, workers change their behaviours depending on density, thereby contact frequency does not linearly increase with density [35].

In this computational study, we show that even at a constant individual density per nest space, colony size dependent behaviours both in queens and workers emerged. This demonstrates that the behavioural changes caused by the feedback loop (which couples the internal state of the queen and workers) code the information regarding the contact rate of the individual worker by the queen. Note that in our simulations all the agents are assumed to exhibit a random walk, i.e., no grouping or clustering, in a constant individual density per space. This demonstrates that it is not the simple overall frequency of encounters, but instead, the two types of specific contact rates that play a role; the contact rate of the queen with reproductive workers and the contact rate of the worker with the queen. The former contact rate is a measure of the inverse of how completely the queen can make contact with workers. The latter is how often individual workers can be contacted by the queen. Due to the contact rate of the individual worker decreasing with colony size, the internal state of the worker increases. Through the resulting change in the internal states, the queen's patrol behaviour is controlled as if she perceives the colony size as discussed previously. Furthermore, the queen patrol efficiency decreases in very large colonies presumably due to some constraints (see later), which leads to the colony stage shifting from the ergonomic stage to the reproductive one, a general phenomenon considered to be adaptive. This discovery is quite novel in that it reveals a single real-time feedback system can control both suppression of worker reproduction in small colonies and its release in large colonies. In monogynous colonies when the queen pheromone is transmitted by direct physical contact between the queen and workers, we consider that this mechanism can generally operate. When these situations arise, the queen-to-worker ratio in the group can be of key importance.

Now we consider the generality of the model presented in this paper in relation to both *Diacamma cf. Indicum* and other social insects. As we have shown, our model is able to replicate the patrol behaviour observed in the queen for *Diacamma cf. Indicum*, with increases in the patrol frequency as a function of colony size. Additionally, previous work [24] has shown the colony size distribution for *Diacamma* with most colonies containing less than 120 workers.

From our results (Fig 6), we show that there is a transition between $N = 80$ and $N = 120$ where the queen's control on worker internal state weakens, with an increase in the internal state of workers. Thus, our model adds value in its explanation of the field observations of real *Diacamma* colonies. With regards to other social insects, the applicability of our model is dependant on the way information is transmitted across the colony. For Bumblebees and Honeybees [24] where queen presence is transmitted through low volatility CHCs, our model could be relevant and adapted to investigate the effectiveness of the queen presence in those colonies and the suppression of worker reproduction. Other eusocial insects such as *Pachycondyla* and *Dinoponera* use dominance interactions from the queen to control worker reproduction in the colony [59, 60]. For such insects, our model could be applicable as a mechanism for the enforcement of the reproductive division of labour. However, for social insects with much larger colonies (such as leaf cutter ants) it would likely be impractical given decreases in queen patrol effectiveness shown in this paper. Queen patrol behaviour would have to be observed in such colonies and other mechanisms would have to be taken into account when determining the importance of such a behaviour in the dynamics of the colony.

## Conclusion

A key assumption of our real-time feedback model is that the queen can perceive the reproductive status (an internal physiological state) of a worker when she contacts it. More importantly, the model also assumes that contact with a reproductive worker(s) leads to an increase in the frequency of queen patrols. These are, however, necessary to empirically demonstrate in experiments using *Diacamma*.

There is another issue that remains to be addressed. Why should the queen's patrol behaviour peak at a certain rate in real *Diacamma* colonies, even if the colony size expands further? The peak queen patrol time is only 20% to 30% of the total time available (S1 Fig), and thus, in principle, the queen could afford to increase her patrol effort further. If queens could significantly increase the frequency of their patrol behaviour, the suppression of worker reproduction would be achieved even in large colonies. To understand the adaptive strategies of queens, we must clarify the limiting factor of the queen's effort investment in patrolling large colonies. One hypothesis is that excessive investment in patrolling might have some fitness costs such as diminished survival and fecundity, which should also be empirically studied in the future.

Also, as to proximate mechanisms of reproductive division of labour in *Diacamma*, we have to take into account other mechanisms, such as dominance behaviour between workers and worker policing. Dominance behaviour is a worker-worker aggressive interaction over the right to produce own male offspring, which occurs both in the absence of the queen [26] and in the presence of the queen, and finally forms a linear hierarchy among workers [61, 62]. Interestingly, similar to the patrol behaviour of queens, ritualised aggressive behaviours by dominant individuals can have an inhibitory effect on the reproductive physiology of subordinate workers [33]. The frequency of dominance behaviours is known to increase with colony size in queen-right colonies [24], which might have a complementary effect to suppress worker reproduction when the efficiency of queen patrol declines. Worker policing, destruction of worker-produced eggs and aggression to an ovary-developed worker by other workers exist in *Diacamma* [32, 51], of which occurrence is also colony-size dependent [50]. Future studies need to develop a simulation model that involves these two mechanisms simultaneously operating. We believe that future research directions discussed above will further enhance our understanding of the mechanisms of the reproductive division of labour in social insects.

## Appendix 1

The frequency of patrols in 12h was positively associated with the colony size (GLMM, $\chi^2$ = 9.396, $P$ = 0.002, S1A Fig). The mean resting time (time between two serial patrols) was negatively correlated with the colony size ($\chi^2$ = 11.202, $P$ = 0.0008, S1B Fig). This finding confirms the results of Kikuchi *et al.* [24].

## Appendix 2

We focused on the first 20 patrol bouts for each queen. The mean patrol duration was 40.6 ± 36.0 sec (SD), and each queen contacted on average 13.1 workers per patrol. The longer the patrol duration, the more workers were encountered during the patrol (GLMM, $\chi^2$ = 475.42, P < 0.001). However, the mean patrol duration was not significantly correlated with colony size (GLMM, $\chi^2$ = 1.148, $P$ = 0.264). The cumulative percentage of workers that a queen encountered in 20 patrols was negatively associated with colony size (GLM, colony size: $z$ = −5.93, $P$ < 0.001, S2 Fig).

## Appendix 3

Finally, we analysed the spatial distribution of workers and the queen, because we had an impression that worker density in the vicinity of the queen is regulated to be relatively constant. Note that we provided an artificial nest of the same design to all 15 colonies. The nest space (the depression of the plaster floor) seemed wide enough for even the largest colony containing 214 workers. Inside the nest, workers tended to aggregate around the queen. Within such an aggregation, spacing between workers seemed more or less constant irrespective of the colony size: in large colonies, a wide space within the nest was occupied by such an aggregation, whereas in small colonies, the aggregation used only a small portion of the nest space (S3 Fig).

## Appendix 4

To test this observation statistically, using the video data for the 15 colonies of *Diacamma*, we made a snapshot of the inside of a nest every 2h, for a total of five times for each colony. We counted the number of workers inside the circle of 2.5-cm radius, the centre of which was positioned on the petiole of the queen. We only counted workers who had over 50% of the body area inside the circle. We excluded snapshot data in which the queen stayed near the wall (within 2.5-cm). The worker density in the circle was not significantly correlated with the colony size (GLMM, $\chi^2$ = 1.302, $P$ = 0.24354, S4 Fig), suggesting that the local worker density around the queen was kept roughly constant regardless of colony size (mean ± SD: 7.04 ± 1.89 workers). Thus, for the queen, a simple encounter frequency with workers is not a reliable proxy of the colony size.

## Appendix 5

Using data we collected from the agent-based simulation, we plotted the probability distribution of worker internal states for a larger colony size ($N$ = 120) to observe the effect of colony size on the distribution over time. Our real-time feedback was effective in controlling the internal state of workers over time (S5 Fig). With no feedback, workers' internal states simply increase. Although, at the larger colony size, there is an increase in the mean internal state and the variance of the distribution. This reflects a weakening of the control the queen has on the reproduction of workers at larger colony sizes as opposed to smaller colonies.

### Appendix 6

Previous work of Kikuchi *et al.* [24] found that while the rest time of the queen decreased with colony size, the patrol time did not seem to significantly increase. We reflected this in the simulation by setting the patrol time of the queen to be constant. However, as shown in Fig 4A, the patrol frequency of the queen increases with colony size. S6 Fig shows the queen's activity cycle for $N = 20$ and $N = 200$. In the simulation code, when the queen is active (and therefore patrolling) the variable "QueenActive" is set to 1, otherwise, it is set to 0. Though the active time of the queen stays the same, the decrease in the rest time causes shorter delays between each patrol when the colony size is large. This is demonstrated in S6 Fig where the increased closeness of the patrols can be seen for $N = 200$.

### Appendix 7

The reason for this increase in patrol frequency is the internal state of the queen. The queen's internal state represents a transition probability. If the queen is inactive and interacts with workers of a high internal state, it increases the probability that the queen will become active and patrol the colony. S7 Fig shows the queen's internal state over time for colony sizes $N = 20$, 100 and 200. As the colony size increases, the internal state of the queen also increases, with a similar trend for colony sizes. There are periods where the queen's internal state is high, followed by significant drops. The larger colony size means that there are more workers to interact with and patrol. From Fig 6 we see that higher colony sizes have a higher average internal state for the workers. This is reflected in the queen also as a higher internal state as she interacts with workers which have an average higher internal state in larger colonies.

### Appendix 8

To check that the dynamics are not influenced by the initialisation of the internal state of the queen, we also ran simulations where the internal state of the queen was initialised at 0.8 (instead of the default 0.1). S8 Fig shows that, despite a different initial value, the queen's internal state mirrors the trends shown in the previous figure, with an increase in the queens internal state as the colony size increases. We also compared directly the internal state of the queen at $N = 20$ when the initial value was 0.1 and 0.8 (S9 Fig). This confirmed that the queen's internal state returns to similar values despite the increase in the initial value showing the robustness of the dynamics of the system.

### Appendix 9

To further confirm our results, we changed the end criteria of the simulation. Initially, the simulation would end when the queen had contacted all workers in a colony. The time it took to accomplish this differed between colony sizes. Thus, the results may not be reflective of the system at a steady state. To account for this, we simulated 300 timesteps across all colony sizes (shown in S10 Fig). We found that the effectiveness of the queen's patrol is strengthened. While there is still an increase in the average internal state of workers as the colony size increases, the suppression of the internal state of workers continues for larger colony sizes than what is shown in Fig 6. This means that our results may underestimate the effectiveness of the queen's patrol behaviour.

### Appendix 10

However, the effectiveness of the queen's patrol behaviour is also linked to how the workers develop their internal state. What would happen if workers developed their internal state more

rapidly? To answer this, we increased the constants $\beta$ and $\gamma$. From Eq 3, $\beta$ and $\gamma$ control the rate at which workers develop their internal state. By increasing their value to 10x the original value, S11 Fig showed that the effectiveness of the queen's patrol behaviour in this model is reliant on the rate that workers develop their internal state. The dynamics are similar, with weaker control as the colony size increases, but total loss of control occurs at smaller colony sizes. In the main results $\beta$ and $\gamma$ were set to approximate the development rate found in the previous work of Kikuchi *et al.* [24].

## Supporting information

**S1 Fig. Changes in the (A) frequency of patrol behaviour and (B) mean resting time of the queen at various colony sizes (N = 3).**
(TIF)

**S2 Fig. Proportion of workers contacted by the queen at least once in 20 patrol bouts at various colony sizes (N = 15).**
(TIF)

**S3 Fig. Aggregation patterns of *Diacamma* individuals at various colony sizes in the artificial nest.**
(TIF)

**S4 Fig. Density of workers within a 2.5-cm radius of the queen at various colony sizes (N = 15).**
(TIF)

**S5 Fig. Probability distribution of worker's internal state (N = 120).** (Top) The probability distribution of workers' internal states over time in a colony with 120 workers with (red) and without (blue) real-time feedback. The average value increases from 0.2473 to 0.2888. (Bottom) The variance is greater at this colony size. This shows a decrease in the effectiveness of the queen's patrol behaviour at larger colony sizes.
(TIFF)

**S6 Fig. Activity cycle of the queen at N = 20 and N = 200.** The activity cycle of the queen changes with colony size. As the colony size increases, the rest time of the queen decreases. This increases the frequency of patrol for the queen at larger colony sizes. Here this is seen as clusters of blue lines, with more clusters when $N = 200$.
(TIF)

**S7 Fig. Queen internal state over timestep.** The queen's internal state changes over time and is coupled with the internal state of workers. Workers in a larger colony have a higher average internal state, causing the internal state of the queen to increase with colony size.
(TIF)

**S8 Fig. Altered queen internal state with higher initialised value.** The initial value of the queen's internal state does not affect the dynamics of the system. Given a higher initial value, the queen's internal state returns to the normal range observed in the previous figure, with larger colony sizes causing an increased internal state as before.
(TIF)

**S9 Fig. Comparison between different initialised value.** Comparing the internal state of the queen for the same colony size with different initial values, we find that there is a convergence

in the queen's internal state after approximately 400 time steps. This shows that the initialisation of the queen's internal state does not affect the dynamics.
(TIF)

**S10 Fig. Running simulation for longer period of time.** By running the simulation for a consistent period of time for each colony size, we guarantee a steady state for each. With this, we see greater control by the queen over the internal state of workers. There is still weakening in the effectiveness of the queen's patrol behaviour but the reversal of the suppression occurs much later at the largest colony sizes.
(TIF)

**S11 Fig. Increasing $\beta$ and $\gamma$.** By increasing $\beta$ and $\gamma$, the rate that workers develop their internal state, we showed that the queen has weaker control over the reproduction of workers. Loss of control begins even at smaller colony sizes such as $N = 40$.
(TIF)

## Acknowledgments

We would like to thank Melissa Winder for assisting in running the simulations. We would also like to thank Oliver Back for his feedback on improving the clarity of the writing. We would like to thank Yuka Fujito, Nao Fujiwara-Tsuji, Shun-ichi Kawabata and Ryohei Yamaoka for discussions that helped to shape the flow of the text. We appreciate Ken Sugawara for his discussion of the work he did which provided a basis for this paper. We are grateful to Toshiharu Akino for helping with a behavioural bioassay, and to Ryo Hosomi and Nao Shigenari who collected preliminary data in their graduation theses at Toyama University.

## Author Contributions

**Conceptualization:** Kazuki Tsuji, Ken Sugawara, Yoshikatsu Hayashi.

**Data curation:** Simeon Adejumo, Kana Maruyama-Onda.

**Formal analysis:** Simeon Adejumo, Tomonori Kikuchi.

**Investigation:** Simeon Adejumo, Tomonori Kikuchi, Kana Maruyama-Onda.

**Methodology:** Simeon Adejumo, Tomonori Kikuchi, Kazuki Tsuji, Yoshikatsu Hayashi.

**Software:** Simeon Adejumo.

**Supervision:** Yoshikatsu Hayashi.

**Writing – original draft:** Simeon Adejumo, Tomonori Kikuchi, Kazuki Tsuji, Yoshikatsu Hayashi.

**Writing – review & editing:** Simeon Adejumo, Tomonori Kikuchi.

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
