## [Decision Letter · Decision Letter 0]

26 Aug 2022

Dear Mr Adejumo,

Thank you very much for submitting your manuscript "A real-time feedback system stabilises the regulation of worker reproduction under various colony sizes" for consideration at PLOS Computational Biology.

As with all papers reviewed by the journal, your manuscript was reviewed by members of the editorial board and by several independent reviewers. 

In this case, the paper has been seen by three expert reviewers. They all agree that the model yields interesting results. However, they also raise some serious concerns. In particular, it remains unclear whether some crucial model assumptions are in fact valid. Moreover,  the general exposition needs to be improved. All reviewers provide  constructive feedback on how to do so (below this email). 

To summarize, it seems like it would take quite some work to address these concerns satisfactorily -- this is a 'Major revision' with the stress being on 'Major'.

We cannot make any decision about publication until we have seen the revised manuscript and your response to the reviewers' comments. Your revised manuscript is also likely to be sent to reviewers for further evaluation.

Sincerely,

Christian Hilbe

Academic Editor

PLOS Computational Biology

Natalia Komarova

Section Editor

PLOS Computational Biology

In the meantime, the paper has been seen by three expert reviewers. They all agree that the model yields interesting results. However, they also raise some serious concerns. In particular, it remains unclear whether some crucial model assumptions are in fact valid. Moreover, also the general exposition needs to be improved. All reviewers provide extremely constructive feedback on how to do so.

It seems to me that it would take quite some work to address these concerns satisfactorily (this is a 'Major revision' with the stress being on 'Major'). However, if the authors are willing to do this work, I would be willing to reconsider this manuscript.

Reviewer's Responses to Questions

**Comments to the Authors:**

Reviewer #1: SUMMARY

The authors explain a new mechanism through which the queen of the ant Diacamma regulates her patrolling behaviour as a function of the colony size. The behaviour is implemented in a multi-agent simulator and the simulation results show a decrease in patrolling efficiency as the colony size increases. This pattern is in agreement with field observations and thus judged correct by the authors.

COMMENTS

The investigated problem is interesting and multi-agent simulation is a suitable research methodology for this study. The topic is relevant to the journal readership.

However, the paper contribution, the methods, and the results are not clearly presented. The

== Issues with the main contribution ==

The authors present a new mechanism that they indicate to be superior to the mechanism presented in previous literature. The previous method consisted of the queen regulating her patrolling behaviour as a function of the ant density. The authors say that when the colony size increases, the ant density remains constant, therefore it cannot be used by the queen to regulate her behaviour. However, I do not see how the new mechanism will produce results qualitatively different from the old mechanism.

The new mechanism shows that when the colony size increases, the queen is less effective in patrolling, thus leading to a phase transition of the colony where some workers become fertile. This is the only result that the authors use to indicate that their method is correct.

I would expect the ‘old’ mechanism to lead to the same qualitative results, in fact, when the queen regulates her patrolling behaviour as a function of ant density and the density is constant, then the queen will put constant patrolling. When the colony size increases, the constant patrolling (which was effective in small colonies) will now become less effective leading to the same phase transition.

The authors need to provide a clear comparison between the previous mechanisms and the new mechanism and must bring convincing arguments to explain why one is better than the other. This comparison is not present and the arguments pushed forward are not convincing.

== Issues with the methods ==

Each simulation runs for the time necessary for the queen to encounter once all the workers. As the simulation can be relatively short, especially for small-sized colonies, and it is far from equilibrium, the initialisation of the parameters can have a particularly relevant impact on the results. I suggest the authors study the system at equilibrium, i.e. to run the simulations for a longer time and/or make a more principled initialisation of the parameters. In particular, my intuition is that the initialisation value of the parameter I_q can have an important effect on the dynamics.

The details of the random walk are not presented, however, the type of selected random walk can have decisive effects on the results. I suggest the authors indicate with clarity what is the tested random walk behaviour and that they also test other random walks.

It is unclear what the authors mean when they say on Line 224 - “The queen will not make a contact with the same worker twice in a row.”

If two workers are in a range of 5 units with the queen for some number of timesteps would it mean that the contact alternates between the two workers? Because the queen does not make a contact two times with the same ant in a row but at each step with a different one alternating between the same two ants.

Or, instead, do the authors mean that the queen never has a second contact with the same worker throughout a simulation? In the former case, it is not clear what is the rationale behind such a choice, and the latter case will have a big impact on the internal state of the queen and it does not seem a correct design choice. Please explain better what you implemented and why you made that implementation choice.

== Issues with the presentation ==

The multi-agent simulations are not sufficiently clearly presented.

* In Figure 2, it is not clear what the blue circles are; the colour legend shows that states are represented with colours from green to brown but several blue circles are displayed. It is also not possible to understand how the runs will evolve over time. Is the movement of all agents synchronous or asynchronous? What’s their motion and interaction pattern?

* Figure 3 is missing. I suggest the authors carefully double-check their submission files before sending them to review.

* Section Results - Lines 258-265 - This part of the text pertains to the Methods and not to the Results (and the information was in part already presented earlier)

* Section Results - Lines 268-273 - This part of the text is difficult to read and it is not clear what is the difference between CRQW and CRWQ. How is the distinction between the two done? and why is it important? Additionally, the text is not correctly organised, as the concepts of contact rates are introduced here and then ignore for a few pages and then the results about them appear much later in the text. This description should go just before the presentation of the results of Fig. 7.

* Fig. 4 - Unclear what patrol frequency is. Never defined. Please define all metrics clearly in a way that your work is reproducible.

* Fig. 5 is hard to read and not well presented. Rather than a 3D plot which is hard to visualise and understand, I suggest the authors use a 2D colourmap (with colour shades indicating the values of the PDF)

* Fig. 6 - “... before stabilising at a higher value.” - Higher than what?

* Fig. 7 - “during the rest cycles of the simulation.” – It is unclear why the results are presented in distinct groups in relation to the rest cycles. What does this distinction mean? Resting cycles of whom? the queen or the worker? Please specify clearly how you computed these values. It is not clear how this data is extracted from the simulations, which quantity do you report? How do you compute CRQW and CRWQ, is it the same quantity divided by the colony size? If yes, I would use a better name (e.g. CR total and per worker).

Reviewer #2: Summary

In this paper, the authors offer a model to help understand and explain how gamergate workers of the ant Diacamma can suppress ovarian development and reproduction of workers through direct contact signalling of queen presence/reproductive status. I have been asked to focus more on the biology than the model itself, though I thought the model seemed logical and sound. On the biology side, I think the paper could be edited to improve clarity – although primarily a computational paper, the work is primarily of interest to social insect researchers, and so I think it should be as accessible as possible to those without a computational background. I’ve made some specific suggestions below, but I think there is greater need to explain certain assumptions and parameters, and the structure could be improved as often some piece of important biological or computational information that would help to understand the model came much later in the paper. I think two major improvements would be incorporating more of the real data in the within paper methods with more explanation on the biology of the ant, rather than this just being in the supplement/methods section. Secondly, related to this, more thought could be given to the generality of the model and the ant species it is replicating. Given that Diacamma are highly unique in that they lack a true queen, colonies are relatively small, and worker reproduction in larger colonies seems somewhat common (though how common I don’t think was mentioned), it would be good to see some discussion on whether it is likely the model described here could be applied to other social insect colonies. For example, I don’t think it would work in something like a leaf cutter ant or Lasius colony, where it would be simply impossible for the queen to contact every individual; moreover, we know they do not perform such patrolling behaviour. However, I could see it being more important in something like a bumblebee colony where the queens actively aggress and police reproductive workers. Finally, as the authors point out, the entire model rests on the assumption queens can detect the internal state of worker, but no attempt to support this by referring to other social insect taxa has been made. Given this is such a major assumption I think it critical this limitation be addressed in the introduction and some justification given.

Specific comments

Italicisation of species name throughout.

Line 23: I think the argument here is that even with chemical signalling, there can potentially be an issue of control as the colony increases in size. If so, I think this could be made clearer as it’s the pivotal point the paper rests on. You could also highlight how important this is, i.e., in very large colonies where they may be millions of insects – how is worker suppression achieved?

Line 34: In either case, there are inclusive fitness benefits that drive the evolution of the behaviour.

Line 36: Rather the ovaries begin to develop.

Line 41: Unclear how 9-ODA connects then with the next sentence, perhaps point out that 9-ODA is more volatile, but that a class of highly conserved, low volatility CHCs seem to play the pivotal role of ovary suppression in social Hymenoptera and an analogous solution has been found in termites.

Line 44: Better to say “low volatility” CHCs.

Line 50: I feel like everything above is a bit repetitive and laboured and could be more succinctly put together.

Line 54: Could point out that she is not a real queen but a gamergate worker, but that you henceforth refer to her as such.

Line 75: Perhaps could be more specific – is she aggressive?

Line 174: Unclear here if “control the length of the resting (i.e., inactive) period depending on the level of her internal states” refers to the queen or the worker. I assume the worker but if queen then I am not sure if her internal state changes? Edit: I now understand how the queen’s state changes but this needs to be made clearer.

Line 178: Does the active-inactive cycles of workers mean they are also moving around the grid? And they were determined a priori, but what were they based on? Edit: I see now they do move, but again this needs to be clearer sooner.

Line 182: Presumably this is unique to the relatively small colonies and would not be feasible in massive colonies, especially where workers do not come into contact with the queen once they switch from inside to outside of nest tasks? I see now from the methods that the queen only contacts about 80% of the colony – would it make more sense given you know this constraint to update the model so it stops once she has contacted 80% rather than 100?

Line 190: Again, I’m a bit unclear what is changing about the queen’s internal state, is this from inactive to patrolling or some other (perhaps hormone) level? Edit: again, realise now what is changing but needs to be clearer sooner.

Line 211: It would be really nice if this section could be given some biological context, i.e. how much of the model here is assumed and how much is based on approximating the biology of the ants. To ensure that the paper is understandable to social insect researchers that are not familiar with such mathematic notations, placing these into biological context would be really helpful, e.g. what is the damping factor attempting to replicate in in the real biological system? I think this is where you could incorporate more about the real-world data you collected and how it has informed your model.

Line 216: In our experience the distribution of ants is not random but concentrated around the queen or brood piles, particularly in small colonies (Casillas- Pérez et al. 2021) – how might such a distribution affect your results, since it should make it easier for the queen to contact as many workers as possible?

Line 220: I appreciate the need for some simplification, but it might worth pointing out here or later than in a real colony ants do not move randomly and there will be overlap, which could impact the dynamics of the queens patrol.

Line 226: Here it would be better to say in plainer terms what happens to the internal states so non-computational biologists can follow along. I presume this is simply that the workers internal state resets to 0 if they contact the queen but it is less clear to me what happens to the queen’s internal state. My interpretation of the damping and activation factors is that if she contacts a low internal state worker, there’s a damping of her internal state which lowers the need to patrol to the point patrolling might switch off with enough contacts, but if she contacts a high internal state worker, she increases her need to control and so patrolling continues? Edit: I now understand from the results that the internal state of the queen is how long she will rest for, and that the patrol length is the same, but this is not clear here where it needs to be.

Line 251: This took me by surprise as I don’t think it is mentioned earlier that you also studied real ant colonies. Could you mention this in the methods above very briefly, so it is easier to follow?

Line 252: I think calling the patrols iterations is a little confusing as it sounds like you might be referring to the model, better to call them bouts as in the methods.

Line 261: Okay so this explanation is what I needed above when I was trying to understand the equations and it would make much more sense if it were to come in the methods rather than the results.

Line 276: To clarify, is this result because the internal state of the workers increases with time? This is mentioned in the equation but could be spelled out more clearly.

Line 291: Could be framed as when contact with the queen has no impact on worker state as a control

Line 299: should add “relative to smaller colonies”, and perhaps give mean values? Looks something 0.15 vs. 0.3.

Line 301: I find figure 6 much easier to read and wonder if figure 5 and S5 are required, as I had to really study these to see the pattern? The same style of figure could be made to show the “without feedback” results for all colony sizes and included within figure 6 or separately in the supplement, with just results reported in the main text.

Line 326: I find this section on contact rates difficult to follow as graph 7 is a bit of a challenge initially to read. It might be helpful to keep the y-axis the same across these graphs, so that the results are more comparable, and to make the dots smaller so the error bars are visible. In general, though, I found I had to look at this figure for a long time to understand it and I’m still not 100% sure I do. My biggest problem is that I don’t understand how there can be contact by the queen during the rest phase of queen. It might also be better to separate out the black and blue results into separate graphs that are on the same row as one another, as the multiple axis, colours and legends make it tricky to follow. I also generally discourage relying on acronyms since it just makes the text harder to follow. Hence, I think this section could be summarised more succinctly by simply saying: as colony size, and thus number of queen patrols, increases, the queen has a higher contact rate per unit time with her workers, whereas the rate that workers contact the queen at decreases. Again, I wonder if it might be simpler to show just Fig 7C, and put A and B in the supplement, or perhaps here you can explain why it is important whether the contacts occur at rest or on patrol.

Line 404: This assumes that ants distribute themselves evenly across nest space though, which I think is unlikely. If a colony concentrates most of its workers in particular chambers or has a stable proportion of workers dedicated to certain task that interact with one another, then encounter rate might be a predictable estimate of colony size.

Line 425: I’m not sure I follow this argument which seems to imply it’s the lack of queen policing that causes the switch, but queen policing efficiency is colinear with increasing colony size, the benefits of which seems like a more general and simply explanation for this shift (i.e., greater brood to worker ratio, more food coming into the nest etc).

Line 443: this assumption is a major one and something that occurred to me throughout the paper. Is there any evidence in this species, other ants, or in other social insects (perhaps in honeybees/ bumblebees) that queens can do this? I also think this needs to be addressed in the introduction as readers will be left wondering about it througought.

Line 446: It might be worth citing Bear et al 2006 or Camargo et al. 2011 which find such trade-offs between an energetic activity (immune response or digging) and fecundity/survival. Though worth noting these are all in claustral systems.

Reviewer #3: See uploaded document.

**Have the authors made all data and (if applicable) computational code underlying the findings in their manuscript fully available?**

Reviewer #1: Yes

Reviewer #2: Yes

Reviewer #3: Yes

PLOS authors have the option to publish the peer review history of their article (what does this mean?). If published, this will include your full peer review and any attached files.

Reviewer #1: No

Reviewer #2: No

Reviewer #3: No
---

## [Decision Letter · Decision Letter 1]

25 Jan 2023

Dear Mr Adejumo,

Thank you very much for submitting your manuscript "A real-time feedback system stabilises the regulation of worker reproduction under various colony sizes" for consideration at PLOS Computational Biology. As noted previously, due to changes to the manuscript file at post-accept your manuscript will need to be sent back to peer-review. In order to do so, we will require you to upload the manuscript file with tracked changes and a new response to reviewers file.

As with all papers reviewed by the journal, your manuscript was reviewed by members of the editorial board and by several independent reviewers. The reviewers appreciated the attention to an important topic. Based on the reviews, we are likely to accept this manuscript for publication, providing that you modify the manuscript according to the review recommendations.

Sincerely,

Christian Hilbe

Academic Editor

PLOS Computational Biology

Natalia Komarova

Section Editor

PLOS Computational Biology

Reviewer's Responses to Questions

**Comments to the Authors:**

Reviewer #1: In their revised manuscript, the authors addressed all my original concerns.

Reviewer #2: The authors have made their manuscript more accessible and justified several points I raised, and I think it is now fit for publication.

Reviewer #3: I consider that, through the revision, almost all issues addressed in my last reviewer’s comments were resolved, and the manuscript deserves publication after minor revisions on the following points,

1. I feel the mechanism for the result shown by the decreasing graph (blue line) in fig.8 is unclear. Why does the per-worker contact rate decrease with the colony size under the assumptions; i) the internal state change of workers does not affect the movement of workers, ii)the number density of workers is kept constant independent of the colony size? Because this result looks (at least for me) against intuition, the authors are desired to suggest some underlying mechanism.

2. This is a small but unignorable issue. In the explanation, in L269-270, of the last terms of eq. (2) and (3), delta(x_q-x_w), authors call this function a “delta function”. This is against the widely accepted definition of the delta function because zero point of the delta function is infinity.

The authors should call this term the “Kronecker’s delta” instead of a “delta function”

and should denote as delta_{x_q, x_w)}.

**Have the authors made all data and (if applicable) computational code underlying the findings in their manuscript fully available?**

Reviewer #1: Yes

Reviewer #2: Yes

Reviewer #3: Yes

PLOS authors have the option to publish the peer review history of their article (what does this mean?). If published, this will include your full peer review and any attached files.

Reviewer #1: No

Reviewer #2: No

Reviewer #3: **Yes: **Hiraku Nishimori

Figure Files:

Data Requirements:

Reproducibility:

References:

---

## [Decision Letter · Decision Letter 2]

10 Feb 2023

Dear Mr Adejumo,

We are pleased to inform you that your manuscript 'A real-time feedback system stabilises the regulation of worker reproduction under various colony sizes' has been provisionally accepted for publication in PLOS Computational Biology.

Best regards,

Christian Hilbe

Academic Editor

PLOS Computational Biology

Natalia Komarova

Section Editor

PLOS Computational Biology

Because the paper has been revised rather substantially, we decided to ask one of the original reviewers to comment on the changes.

As you will see, this reviewer suggested to publish the article, and we concur.

Ideally, you briefly take into account the reviewer's remaining suggestions when uploading the final manuscript.

Reviewer's Responses to Questions

**Comments to the Authors:**

Reviewer #3: Although I consider that the manuscript now deserves publication, a little more precise definition of the term, “per-worker contact rate between workers”, is desired to be added because I still have not completely understood its meaning.

Does the “per-worker contact rate between workers” mean the contact frequency of a worker to a particular worker, or, her contact frequency to any of the other workers in the colony?

I guess that the answer is, most probably, the former, then, I completely understand that the decrease in the per-worker contact rate in fig.8 is explained by the same reason as the decrease in the per-worker contact rate of the queen.

But if the answer is the latter, the per-worker contact rate seems to increase (or is kept unchanged) for the same reason as the increase of the queen contact rate to workers.

In addition, the formal notation of the “Kronecker’s delta” should be “delta_{x_q, x_w}”, rather than “delta_(x_q- x_w)”.

I consider that no further review is needed after these minor revisions.

**Have the authors made all data and (if applicable) computational code underlying the findings in their manuscript fully available?**

Reviewer #3: Yes

PLOS authors have the option to publish the peer review history of their article (what does this mean?). If published, this will include your full peer review and any attached files.

Reviewer #3: **Yes: **Hiraku Nishimori

---

## [Editor Report · Acceptance letter]

21 Mar 2023

PCOMPBIOL-D-22-00963R2 

A real-time feedback system stabilises the regulation of worker reproduction under various colony sizes

Dear Dr Adejumo,

I am pleased to inform you that your manuscript has been formally accepted for publication in PLOS Computational Biology. Your manuscript is now with our production department and you will be notified of the publication date in due course.

With kind regards,

Anita Estes
